# Meropenem Plasma and Interstitial Soft Tissue Concentrations in Obese and Nonobese Patients—A Controlled Clinical Trial

**DOI:** 10.3390/antibiotics9120931

**Published:** 2020-12-21

**Authors:** Philipp Simon, David Petroff, David Busse, Jana Heyne, Felix Girrbach, Arne Dietrich, Alexander Kratzer, Markus Zeitlinger, Charlotte Kloft, Frieder Kees, Hermann Wrigge, Christoph Dorn

**Affiliations:** 1Department of Anaesthesiology and Intensive Care Medicine, University of Leipzig Medical Centre, 04103 Leipzig, Germany; jana.roemuss@gmail.com (J.H.); Felix.Girrbach@medizin.uni-leipzig.de (F.G.); hermann.wrigge@bergmannstrost.de (H.W.); 2Integrated Research and Treatment Center (IFB) Adiposity Diseases, University of Leipzig, 04103 Leipzig, Germany; david.petroff@zks.uni-leipzig.de (D.P.); arne.dietrich@medizin.uni-leipzig.de (A.D.); 3Clinical Trial Centre Leipzig, University of Leipzig, 04107 Leipzig, Germany; 4Department of Clinical Pharmacy and Biochemistry, Institute of Pharmacy, Freie Universitaet Berlin, 12169 Berlin, Germany; dbusse@zedat.fu-berlin.de (D.B.); charlotte.kloft@fu-berlin.de (C.K.); 5Graduate Research Training Program PharMetrX, 12169 Berlin, Germany; 6Department of Surgery, University of Leipzig Medical Centre, 04103 Leipzig, Germany; 7Institute of Pharmacy, University of Regensburg, 93053 Regensburg, Germany; Alexander.Kratzer@klinik.uni-regensburg.de (A.K.); Christoph.Dorn@chemie.uni-regensburg.de (C.D.); 8Department of Clinical Pharmacology, Medical University of Vienna, 1090 Vienna, Austria; markus.zeitlinger@meduniwien.ac.at; 9Department of Pharmacology, University of Regensburg, 93053 Regensburg, Germany; frieder.kees@web.de; 10Department of Anaesthesiology, Intensive Care and Emergency Medicine, Pain Therapy, Bergmannstrost Hospital Halle, 06112 Halle, Germany

**Keywords:** antibiotic dosing, concentrations, meropenem, microdialysis, obesity, pharmacokinetics, soft tissue, pharmacodynamics

## Abstract

Background: This controlled clinical study aimed to investigate the impact of obesity on plasma and tissue pharmacokinetics of meropenem. Methods: Obese (body mass index (BMI) ≥ 35 kg/m^2^) and age-/sex-matched nonobese (18.5 kg/m^2^ ≥ BMI ≤ 30 kg/m^2^) surgical patients received a short-term infusion of 1000-mg meropenem. Concentrations were determined via high performance liquid chromatography-ultraviolet (HPLC-UV) in the plasma and microdialysate from the interstitial fluid (ISF) of subcutaneous tissue up to eight h after dosing. An analysis was performed in the plasma and ISF by noncompartmental methods. Results: The maximum plasma concentrations in 15 obese (BMI 49 ± 11 kg/m^2^) and 15 nonobese (BMI 24 ± 2 kg/m^2^) patients were 54.0 vs. 63.9 mg/L (95% CI for difference: −18.3 to −3.5). The volume of distribution was 22.4 vs. 17.6 L, (2.6–9.1), but the clearance was comparable (12.5 vs. 11.1 L/h, −1.4 to 3.1), leading to a longer half-life (1.52 vs. 1.31 h, 0.05–0.37) and fairly similar area under the curve (AUC)_8h_ (78.7 vs. 89.2 mg*h/L, −21.4 to 8.6). In the ISF, the maximum concentrations differed significantly (12.6 vs. 18.6 L, −16.8 to −0.8) but not the AUC_8h_ (28.5 vs. 42.0 mg*h/L, −33.9 to 5.4). Time above the MIC (T > MIC) in the plasma and ISF did not differ significantly for MICs of 0.25–8 mg/L. Conclusions: In morbidly obese patients, meropenem has lower maximum concentrations and higher volumes of distribution. However, due to the slightly longer half-life, obesity has no influence on the T > MIC, so dose adjustments for obesity seem unnecessary.

## 1. Introduction

Meropenem is a carbapenem with a broad spectrum of activity against Gram-positive and Gram-negative bacteria, including multiresistant pathogens, and is therefore recommended for the calculated therapy of patients with sepsis [1,2]. For the success of an anti-infective therapy and the prevention of resistance development, the choice of the right antibiotic and the right duration of therapy, as well as the correct dosage, play decisive roles. An effective therapy requires sufficiently high antibiotic concentrations at the target site over a sufficient period of time.

Obesity is a growing problem worldwide [3] and constitutes a risk factor for the acquisition of serious infections, as well as for their successful treatment [4]. However, recommended daily doses of antibiotics are based on pharmacokinetic (PK) studies performed almost exclusively in nonobese patients [5]. Data to characterize the pharmacokinetics (PK) in obese compared with nonobese patients are thus increasingly important to assess whether effective meropenem concentrations are reached also in this growing population. Recent studies indicate that the current standard dosage of meropenem in obese patients is sufficient [6,7,8], and there are no recommendations for weight-dependent dosing [9]. However, the majority of these studies use plasma concentrations as the basis for their analyses, but ideally, also, target site (e.g., tissue) interstitial fluid (ISF) concentrations should be included [10,11,12]. Only one study with a limited number of obese patients has investigated the plasma, as well as the tissue PK, of meropenem, but the direct influence of obesity could not be assessed, since no nonobese control group was implemented [13]. A direct comparison of meropenem tissue concentrations in obese vs. nonobese patients is still lacking.

We hypothesized that obesity has an influence on the pharmacokinetics of meropenem in plasma and the ISF of subcutaneous adipose tissue (target site) and investigated whether this might necessitate dose adjustment. To answer this question, we studied obese and nonobese patients undergoing elective abdominal surgery in a controlled setting, measuring meropenem concentrations in the plasma, as well as in subcutaneous tissue, via microdialysis.

## 2. Results

### 2.1. Patients

Fifteen obese (two class II and 13 class III) patients scheduled for bariatric surgery and 15 nonobese patients undergoing elective abdominal surgery, mainly tumour resection (11 patients underwent gynaecological operations, and the remaining four operations involved the stomach, liver, kidneys or appendix), were included in the study (see Figure 1—CONSORT flow diagram). The patient characteristics are presented in Table 1. No patient showed signs of acute infection or critical illness at the time of the surgery and the first seven postoperative days. No drug-related adverse events were observed. No interactions between patients’ medications were found to affect the pharmacokinetics of meropenem.

### 2.2. Pharmacokinetics

The plasma and ISF data were available from all 30 patients. Free plasma concentrations were determined in 69 samples. The unbound fraction (*f*u) was independent of the total concentration and amounted to 98.3% ± 3.9% (95% CI: 97.4–99.3), with a mean intraindividual coefficient of variation of 2.5%. The difference to 100% was not considered to be clinically relevant, and the total concentrations were used for the pharmacokinetic/pharmacodynamic (PK/PD) calculations. The mean (total) plasma concentration–time curve of meropenem was characterized by a lower maximum concentration (C_max_) and a smaller slope of the descending trajectory of the concentration–time profile in obese patients compared with nonobese patients, leading, on average, to an intersection of the curves at about three hours after infusion (Figure 2). 

In two patients, only one catheter was evaluable due to a malfunction of the second catheter. There was no significant difference between the recovery of the catheters in the right and left arms. Therefore, the results of both catheters were averaged. The recovery in obese patients was significantly lower than in the nonobese patients (33.9 vs. 51.2%; 95% CI for the difference was −30.3 to −4.5%, *p* = 0.010). 

The primary endpoint, the area under the concentration–time curve from zero to eight h (AUC_8h_) in the ISF of the subcutaneous adipose tissue, did not differ significantly between obese and nonobese patients, with median values of 28.5 (obese) vs. 42.0 (nonobese) mg/L∙h, with a difference in the location parameter of −13.0 mg/L∙h (95% CI: −33.9 to 5.4 mg/L∙h, *p* = 0.15). The same applied to the AUC extrapolated to infinity (AUC_∞_), with median values of 33.7 (obese) vs. 44.6 (nonobese) mg/L∙h, with a difference in the location parameter of −11.4 mg/L∙h (95% CI −31.9 to 5.5 mg/L∙h, *p* = 0.22). The C_max_ in the ISF were lower in obese compared with nonobese patients, but the concentration curves crossed at about four-to-five h (Figure 2). The pharmacokinetic parameters are summarized in Table 2. 

In the obese group, the C_max_ in the plasma was significantly lower, the volume of distribution at the steady state (V_ss_) higher and the half-life (t_1/2_) longer, whereas no significant differences were found regarding the clearance (CL) or AUC. In the ISF, the results were qualitatively similar, though the difference in the t_1/2_ was not significant, whereas the difference in concentration after eight hours (C_8h_) was. The penetration ratio, expressed as AUC_ISF_/AUC_plasma_, was similar in both groups (median of 0.35 in the obese group and 0.48 in the nonobese group, *p* = 0.40). The V_ss_ was significantly correlated with the weight (R = 0.73, 95% CI: 0.51–0.87, *p* < 0.001, Figure 3). Table 3 shows the T_>MIC_ (time above the minimum inhibitory concentration) values for pathogens with MICs of 0.25, 2 and 8 mg/L; there were no significant differences between the obese and nonobese patients.

## 3. Discussion

The present study is the first clinical study with a direct comparison of meropenem tissue concentration in obese vs. nonobese patients. The summarized findings are that the maximum concentrations in the obese patients were lower than in the nonobese control group, the volume of distribution was higher, the clearance comparable and the half-life longer. As an overall net result, obesity had little or no clinically relevant effect on the PK/PD index T > MIC when evaluating a single short-term infusion of the standard meropenem dose.

The pharmacokinetics of meropenem in humans has been investigated extensively also for critically ill patients [14]. The maximum plasma concentrations we found in the control group agreed well with previous data in healthy volunteers, whereas the half-life was somewhat longer, and the AUC was correspondingly higher [15]. The differences could be due to the older study population, with an age-associated decrease in kidney functions [16]. Further, physiological changes during surgery/anesthesia, such as changes in cardiac output, blood volume, regional blood flow and extracellular fluid volume, may have influenced the elimination of meropenem [17]. 

The main finding of the study is that, in direct comparison with nonobese surgical patients, obese surgical patients have lower maximum concentrations due to a higher volume of distribution but a similar clearance and, thus, a longer half-life. This might have important implications on alternative dosing regimens, such as prolonged infusions of meropenem. The lower ISF concentrations during surgery in obese patients compared with nonobese patients as observed in the present study are in-line with previously published results and may be explained by a lower subcutaneous adipose tissue blood flow [18]. However, these effects of obesity are different for individual antibiotics [9], which is why these results are only valid for meropenem. For other substances (even quite similar ones, like doripenem), separate results are needed for evidence-based recommendations.

All PK/PD indices should be referenced to the (nonprotein-bound) fraction (*f*u) of the drug [19]. For meropenem, the total or free concentrations are equally suitable for calculating the PK/PD parameters, given the nearly absent plasma protein binding. The *f*u of 98% as determined in the present study is commonly accepted as reliable, despite contrasting results in a very recent study reporting a median *f*u of 62% (range 42–99%) only [19,20]. However, except that these results are in contradiction to other studies, no information is provided regarding the validation data of the assay, including the ultrafiltration procedure, as well as the possibility of the preanalytical degradation of meropenem [20,21,22]. 

The decisive PK/PD parameter for beta-lactams is T_>MIC_. The plasma and tissue concentrations remain above the European Committee on Antimicrobial Susceptibility Testing (EUCAST) breakpoints of 2 mg/L for *Pseudomonas* and *Enterobacteriaceae* [23] for comparably long periods of time, suggesting that body weight-adjusted dosages of meropenem in obesity are unnecessary after short-term infusions. In obese patients, the mean values of the T_>MIC_ were somewhat higher in most cases, though the difference was not clinically relevant, because a dose reduction for obese patients would not benefit them clinically. Although the sample size was based on the primary endpoint, the width of the 95% CI suggests that it was also adequate for the T_>MIC._ At first glance, the longer T_>MIC_ in obese patients is a surprising message, considering the lower maximum plasma concentrations in the obese group, but understandable in the context of the time-dependent and not concentration-dependent antibacterial activity of beta-lactams. Due to the longer half-life of meropenem though, the concentrations were equal or even higher in the obese patients. The conclusions of this study are consistent with most previous studies [5,6,7,8,9,13,24], but our study included the pharmacokinetics in the plasma and tissue concentrations, as well as a comparison between an age-matched nonobese control group, thereby placing these conclusions on much firmer ground. 

However, this study had some limitations. The study was performed in surgical noninfected patients under defined conditions. Meropenem is particularly indicated for severe infections and in intensive care unit (ICU) patients, where confounding factors such as changes in the volume of distribution, reduced microcirculation or edema exist. The body mass index (BMI) values in the obese group are very high, and as such, the results may not be entirely applicable in nonbariatric cohorts. Moreover, by design, patients with BMI 30 to 34.9 kg/m^2^ were not included to have a clearer separation between the groups. Since the obese patients did not show any relevant differences to nonobese patients, it can be assumed that these limitations are minor. Furthermore, surgery and/or anesthesia may have affected the distribution and elimination rate of meropenem. This could only be investigated with a control group without surgery/anesthesia and should be investigated in clinical trials in the future.

## 4. Materials and Methods 

### 4.1. Study Design and Patients

This prospective, parallel group, open-label, monocentric, controlled trial was embedded in a larger pharmacokinetic study at the Leipzig University Hospital and was registered in the European Union clinical trials register (EudraCT No. 2012-004383-22) and German Clinical trials Register (DRKS00004776). Previous analyses on linezolid and fosfomycin have been already published [25,26,27]. Approval for the trial was obtained from the Leipzig University Ethics Committee (121/13-28012013) and the Federal Institute for Drugs and Medical Devices of Germany (BfArM—No.: 4038808). Methods, design and sample size calculations were described in detail [28]. Patients scheduled for elective abdominal surgery at Leipzig University Hospital were screened for eligibility. Inclusion criteria comprised ages ≥ 18 years and body mass index (BMI) ≥ 35 kg/m^2^ for obese patients or between 18.5 and 30 kg/m^2^ for nonobese patients. Main exclusion criteria were pregnancy or breastfeeding, known allergic reactions to one of the investigated medications or severe liver or kidney diseases. Prior written informed consent was obtained from all study participants.

### 4.2. Study Procedure

Microdialysis catheters (CMA 63 microdialysis probe, cut-off 20,000 Da, CMA, Kista, Sweden) were placed in the subcutaneous adipose tissue of both upper arms without using local anesthesia (one in each arm) 90 min before beginning general anesthesia and administration of the study drugs. As a perioperative antibiotic prophylaxis, patients were given a standard (weight-independent) dose of 1000-mg meropenem (Meronem^®^, AstraZeneca GmbH, Wedel, Germany) in combination with 600-mg linezolid (Zyvoxid^®^, Pfizer Deutschland GmbH, Berlin, Germany) as a single 30-min infusion after the induction of anesthesia (60–30 min prior to incision) through an additional vein access. An analysis of the linezolid concentrations in these patients was already published separately [25,26] due to the significantly different properties between the two antibiotics. Anesthesia was performed according to clinical standards and left at the discretion of the anesthetist. The catheters were constantly perfused with 0.9% saline at 2 µL/min. Blood samples were taken pre-dose and after 0.5 (end of infusion), 1, 2, 3, 4, 5, 6 and 8 h. Microdialysate samples were collected pre-dose and after dosing at 0 to 0.5, 0.5 to 1, 1 to 1.5, 1.5 to 2, 2 to 3, 3 to 4, 4 to 5, 5 to 6, 6 to 7 and 7 to 8 h. For calibration of the probes, the recovery of each probe was determined by retrodialysis, with meropenem 20 mg/L in saline [28,29]. Retroperfusate (C_RP_) and retrodialysate (C_RD_) drug concentrations were utilized to determine the relative recovery (relative recovery = 1 − (C_RD_/C_RP_) × 100%). The ISF concentrations were then determined by C_ISF_ = C_microdialysate_ × 100/(relative recovery). Blood samples were centrifuged within 30 min of sampling (3000× *g*, 4 °C, 10 min). Plasma and microdialysate samples were frozen immediately at −25 °C and then stored at −80 °C until analysis. Patients were visited preoperatively, intraoperatively and daily between postoperative days 1 and 7 or until discharge, as described in the complete participant timeline [28]. Study data were collected and managed using an Oracle-DBMS database with the eResearch Network. Electronic data capture tools are hosted at the Clinical Trials Centre of the University of Leipzig, Germany.

### 4.3. Drug Analysis

Meropenem concentrations were analyzed adapting a validated high performance liquid chromatography-ultraviolet (HPLC-UV) method [30]. Sample treatment of plasma, followed a published protocol for beta-lactams [31]. In brief, plasma (100 µL) was mixed with 25-mM sodium dihydrogen phosphate (200 µL) and acetonitrile (500 µL). After separation of the precipitated proteins, the acetonitrile was extracted into dichloromethane (1.3 mL). Sample treatment for the determination of the free plasma concentration was performed by ultrafiltration. Plasma (300 µL) was buffered with 3-M potassium phosphate, pH 7.4 (10 µL), incubated (10 min/37 °C/1000× *g*, centrifuge 5417R, Eppendorf, Hamburg, Germany) and then centrifuged (20 min/37 °C/1000× *g*) using Vivacon 500 ultrafiltration devices (Sartorius, Göttingen, Germany). Microdialysate was injected directly. The injection volume was 1–3 µL. Chromatography was performed on a Prominence Modular LC-20 HPLC (Shimadzu, Duisburg, Germany) with photometric detection at 300 nm. The autosampler was cooled to 6 °C, and the column temperature was maintained at 40 °C. The analytical column was a Waters XBridge C18 BEH 2.5 µm 50 × 3 mm (Waters, Eschborn, Germany) or a Macherey-Nagel Nucleoshell RP18 2.7 µm 100 × 3 mm (Macherey-Nagel, Düren, Germany). Both columns were preceded by a Macherey-Nagel Nucleoshell RP18 2.7 µm 4 × 3-mm column protection system. The mobile phase was 0.1-M sodium phosphate buffer/acetonitrile 92:8 (*v/v*) and final pH 2.9–3.1. The flow rate was 0.4 mL/min and the retention time of meropenem 2.5–4.5 min (depending on the exact pH and column). Based on in-process quality controls (50, 15 and 0.5 mg/L in plasma or 15, 1.5 and 0.15 mg/L in saline as the surrogate for microdialysate), the mean intra-/interassay imprecision and inaccuracy were <4%. The accuracy of free meropenem cannot be specified, as the protein binding and, accordingly, also, the true free concentration in an individual plasma sample, are not known. The unbound fraction (*f*u) in the pooled plasma from healthy volunteers, analyzed as quality controls with each assay, was 99.5% ± 1.3%. The lower limit of quantification (LLOQ) was 0.1 mg/L in plasma and 0.02 mg/L in saline.

### 4.4. Study Endpoints 

The primary endpoint was the area under the concentration–time curve from 0 to 8 h (AUC_8h_) of meropenem in the ISF. Secondary endpoints were AUC_8h_ in the plasma, maximum concentration (C_max_), half-life (t_1/2_), AUC_ISF_/AUC_plasma_ ratio and other pharmacokinetic/pharmacodynamic (PK/PD) parameters in the ISF and plasma [28]. For the exploratory PK/PD analysis, we used the T_>MIC_ (time above the minimum inhibitory concentration) [15] with MIC values from 0.25 mg/L to 8 mg/L covering the EUCAST breakpoints of clinically relevant bacteria [23].

### 4.5. Pharmacokinetic and Statistical Analysis

The ISF and plasma concentrations were analyzed on a logarithmic scale. Noncompartmental PK analysis (NCA) was carried out using Phoenix WinNonlin 8.1 (Certara, Princeton, NJ, USA). The elimination rate constant λ_z_ was determined by log-linear regression over the last 3 to 4 data points (terminal phase) and began typically by 4 h (plasma) or 4.5 h (ISF). The linear-up/log-down trapezoidal rule was used for calculation of the AUC_0–8h_. Extrapolation of the AUC to infinity (AUC_∞_) was based on the last predicted concentration. The tissue penetration ratio (AUC_ISF_/AUC_plasma_) was calculated using the AUC_∞_. The ISF parameters were first determined separately in each of the two catheters and then averaged. The software R (The R Foundation, Vienna, Austria, version 3.6.3) was used for the data visualization and calculation of statistical tests. Comparison between the groups was made using the Welch *t*-test or the Wilcoxon Mann-Whitney test. In the first case, differences between means were provided with a 95% Wald confidence interval, and in the second, a Hodges-Lehmann estimator with an exact 95% confidence interval was used. *p* < 0.05 was considered statistically significant.

## 5. Conclusions

The pharmacokinetics of meropenem in morbidly obese patients was characterized by lower maximum plasma concentrations and a higher volume of distribution compared with nonobese patients. Due to the slightly longer half-life, the PK/PD parameter T > MIC of meropenem was not negatively affected by obesity, and body weight-adjusted dosing seems unnecessary in obese patients after a short-term infusion. 

## Figures and Tables

**Figure 1 antibiotics-09-00931-f001:**
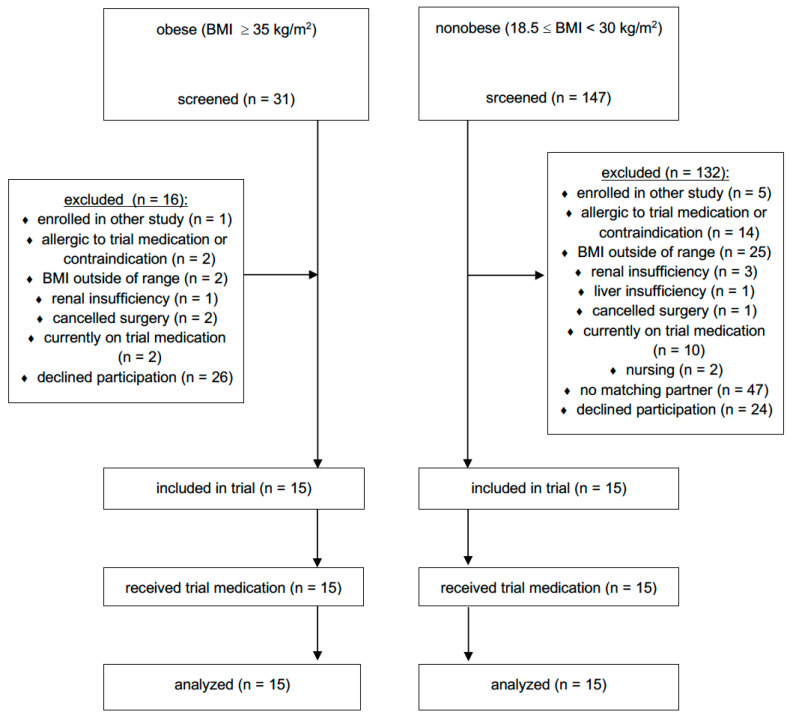
CONSORT flow diagram. BMI = body mass index.

**Figure 2 antibiotics-09-00931-f002:**
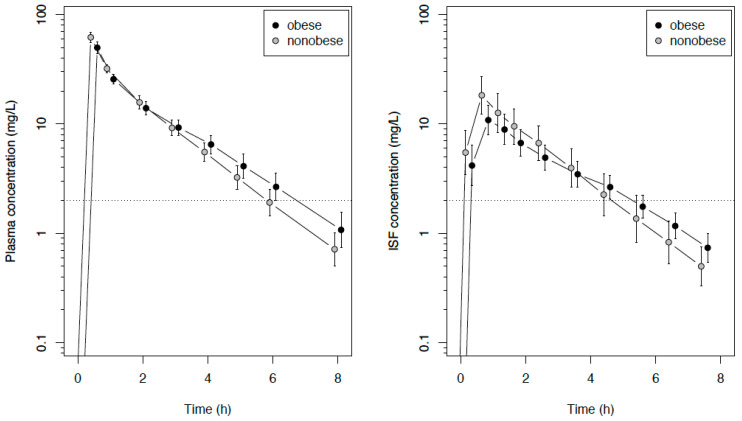
Semilogarithmic concentration–time course (mean and SD) of meropenem in the plasma and interstitial fluid of subcutaneous tissue in obese or nonobese surgical patients following an intravenous infusion of 1 g of meropenem. The dotted line indicates the European Committee on Antimicrobial Susceptibility Testing (EUCAST) breakpoints for *Pseudomonas aeruginosa* and *Enterobacteriaceae.* ISF: interstitial fluid.

**Figure 3 antibiotics-09-00931-f003:**
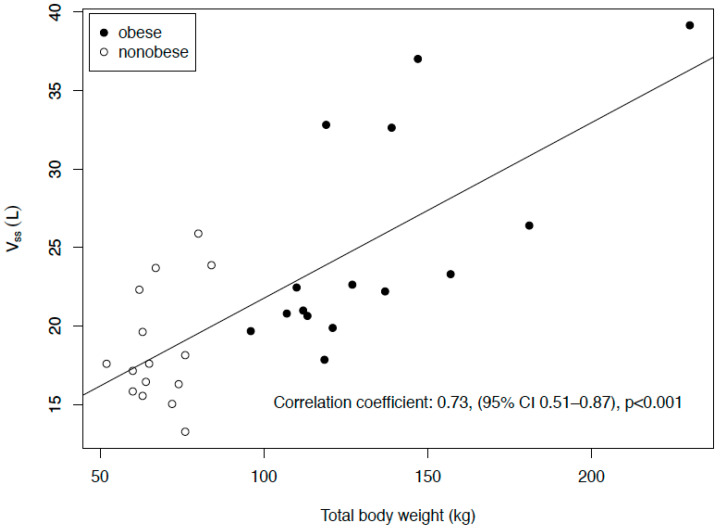
Correlation of the volume of distribution at the steady state (Vss) with the total body weight.

**Table 1 antibiotics-09-00931-t001:** Demographic and clinical characteristics.

	Obese (*n* = 15)	Nonobese (*n* = 15)
Number of females	13 (87%)	13 (87%)
Age (years)	50.3 ± 9.5	49.5 ± 10.0
Weight (kg)	134.3 ± 34.3	67.9 ± 8.8
BMI (kg/m^2^)	48.7 ± 11.2	23.9 ± 2.1
Type of surgery		
laparoscopic	15 (100%)	3 (20%)
open	0 (0%)	12 (80%)
Length of surgery (h)	2.9 ± 0.6	4.0 ± 1.9
Serum creatinine (µmol/L)	85.8 ± 26.3	75.3 ± 18.8

Entries are mean ± standard deviation or numbers (%). BMI = body mass index.

**Table 2 antibiotics-09-00931-t002:** Pharmacokinetic parameters.

Measuerments	Plasma				ISF			
	Obese (*n* = 15)	Nonobese (*n* = 15)	Change in Location Parameter (95% CI)	*p*-Value	Obese (*n* = 15)	Nonobese (*n* = 15)	Change in Location Parameter (95% CI)	*p*-Value
maximum concentration (C_max_ (mg/L))	54.0 (44.8–58.2)	63.9 (53.6–69.9)	−11.0 (−18.3 to −3.5)	0.011	12.6 (6.5–17.1)	18.6 (12.4–31.2)	−7.4 (−16.8 to −0.8)	0.026
Timepoint of maximum (T_max_ (h))	0.5 (end of infusion)	0.5	―	1.0	0.75/1.25 ^a^ (*n* = 12/3)	0.75/1.25 ^a^ (*n* = 14/1)	―	0.60
Concentration after 8 hours (C_8h_ (mg/L))	1.02 (0.74–1.48)	0.70 (0.48–0.88)	0.33 (−0.03 to 0.72)	0.070	0.54 (0.45–0.77) ^b^	0.36 (0.24–0.51) ^b^	0.21 (0.01–0.38)	0.041
Half-life (t_1/2_ (h)) ^c^	1.52 (1.44–1.67)	1.31 (1.23–1.43)	0.22 (0.05 to 0.37)	0.0066	1.55 (1.44–1.66)	1.32 (1.14–1.49)	0.24 (−0.02 to 0.42)	0.061
Volume distribution at steady state (V_ss_ (L))	22.4 (20.7–29.5)	17.6 (16.1–21.0)	5.1 (2.6 to 9.1)	0.0017	―	―	―	―
Clearance (CL(L/h))	12.5 (9.8–13.9)	11.1 (9.8–12.6)	0.8 (−1.4 to 3.1)	0.46	―	―	―	―
Area under the curve 0 to 8 hours (AUC_8h_ (mg * h/L))	78.7 (69.1, 99.6)	89.2 (78.1, 100.9)	−7.2 (−21.4 to 8.6)	0.35	28.5 (19.2, 44.4)	42.0 (29.7, 66.0)	−13.0 (−33.9 to 5.4)	0.15
Area under the curve extrapolated to infinity (AUC_∞_ (mg * h/L)) ^d^	80.3 (71.8–102.0)	90.3 (79.4–102.3)	−6.5 (−21.1 to 10.4)	0.46	33.7 (23.6–38.7)	44.6 (29.2–59.1)	−11.4 (−31.9 to 5.5)	0.22
Penetration ratio (AUC_∞_, ISF/AUC_∞_, plasma)	―	―		―	0.35 (0.27–0.52)	0.48 (0.33–0.62)	−0.09 (−0.30 to 0.08)	0.40

Entries are median (interquartile range). ^a^ Fraction 0.5–1 h and 1–1.5 h, respectively, ^b^ to 8.0-h extrapolated concentration. ^c^ The median adjusted R-squared of the regressions for the terminal phases were 0.997 in the plasma and 0.987 in the interstitial fluid (ISF). ^d^ The median extrapolated aera under the curve (AUC) was 1.8% in the plasma and 5.9% in the ISF.

**Table 3 antibiotics-09-00931-t003:** Time (in hours) for which the concentration is above the minimum inhibitory concentration (MIC) (T > MIC).

	Obese (*n* = 15)		Nonobese (*n* = 15)		Difference between Groups		*p*-Value
	Estimate	95% CI	Estimate	95% CI	Estimate	95% CI	
MIC in Plasma							
0.25 mg/L	11.9	(10.3 to 13.4)	10.2	(9.3 to 11.2)	+1.6	(−0.3 to 3.6)	0.097
2 mg/L	6.9	(6.0 to 7.8)	6.1	(5.5 to 6.8)	+0.8	(−0.3 to 2.0)	0.16
8 mg/L	3.8	(3.3 to 4.2)	3.5	(3.1 to 3.9)	+0.2	(−0.4 to 0.9)	0.42
MIC in ISF							
0.25 mg/L	11.0	(10.1 to 11.8)	9.6	(8.2 to 10.9)	+1.4	(−0.3 to 3.0)	0.10
2 mg/L	5.4	(4.7 to 6.1)	5.0	(4.0 to 6.1)	+0.3	(−1.0 to 1.7)	0.60
8 mg/L ^a^	1.4	(0.7 to 2.1)	2.0	(1.2 to 2.9)	−0.6	(−1.8 to 0.5)	0.26

Entries are mean and 95% CI. ^a^ Five patients from the obese and three from the nonobese group did not reach a MIC (i.e., time above the MIC was zero).

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
