# Peer review of "Meropenem Plasma and Interstitial Soft Tissue Concentrations in Obese and Nonobese Patients—A Controlled Clinical Trial"

_antibiotics, 2020, doi:10.3390/antibiotics9120931_

Round 1
Reviewer 1 Report
The authors of the article took up the important topic of meropenemPK in obese vs healthy patients. Antibiotic PK in obese patients is a fairly well-understood topic. It is a pity that the authors of the study did not analyze the PK of other medications administered orally to patients, where gastrectomy and obesity could reveal more variables. It would be worth supplementing the medications taken by patients. Were the patients taking anticoagulants? has there been any interaction? What are the weaknesses of the study? What are the limitations of the study? What's new in the presented study? Has there been a dose adjustment of the antibiotic in patients? What bacteria were identified in the blood and tissue cultures? Has PK / PD been calculated separately for specific pathogens? In the title of Figure 2, it should be stated that it is a logarithmic figure.
Author Response
Point-by-point response to the reviewers’ comments to antibiotics-104004
“Meropenem plasma and interstitial soft tissue concentrations in obese
and non-obese patients – a controlled clinical trial”
by Simon et al.
Dear Prof. Dr. Nicholas Dixon, Editor-in-Chief,
Dear Dr. David P. Nicolau, Guest Editor,
Dear Leah Zhou,
Thank you very much for giving us the opportunity to revise our manuscript entitled „ Meropenem plasma and interstitial soft tissue concentrations in obese and non-obese patients – a controlled clinical trial”.
We thank the editors and reviewers for their criticism. We have discussed the points in detail and hope very much that after further revision you will come to the conclusion that the manuscript is suitable for publication in Antibiotics.
Our responses to the reviewers’ comments below are highlighted in blue and italic, our changes to the manuscript (insertions and deletions) are indicated in red (deletions are crossed through). The line numbers refer to those in the revised manuscript with highlighted changes.
REVIEWER 1:
- The authors of the article took up the important topic of meropenem PK in obese vs healthy patients. Antibiotic PK in obese patients is a fairly well-understood topic.
Antibiotic PK in obese patients is a topic of growing importance and, indeed, much knowledge has been acquired in the past years. We agree that antibiotic PK in obese patients is a fairly well-understood topic in general, but solid data on specific substances is often missing. Recent studies indicate that the current standard dosage of meropenem is sufficient for obese patients, but most of these studies rely on plasma data only, ignoring the impact of obesity on tissue penetration. Only one study with a limited number of obese patients (2 male, 3 female) has investigated the plasma as well as the tissue PK of meropenem, but the direct influence of obesity could not be assessed since no non-obese control group was implemented [13 in the manuscript]. The present study is the first clinical study with a direct comparison of meropenem tissue concentration in 15 obese vs 15 non-obese patients.
We have added the following sentence in the Introduction (page 2; lines 68-69):
A direct comparison of meropenem tissue concentrations in obese versus non-obese patients is still lacking.
- It is a pity that the authors of the study did not analyze the PK of other medications administered orally to patients, where gastrectomy and obesity could reveal more variables.
Indeed, the impact of gastrectomy and/or obesity on the PK of orally administered drugs is an interesting topic, but was not the purpose of this large clinical microdialysis trial. As pointed out in the previously published article describing the rationale and design of the trial [25 in manuscript], the aim of the trial was to investigate whether current intravenous dosing regimens of various antibiotics lead to effective concentrations in the ISF of morbidly obese patients. When this trial was designed, we also felt that several substances should be investigated once the investment in techniques and logistics was made. The complexity of the topic and specific questions relevant to each substance mean that they do not lend themselves to publication in a single paper. Hence, we have already presented data on linezolid and fosfomycin and describe this more clearly now in the paper.
We have added the following sentence to the methods, to refer to previous results (page 9, line 204):
Previous analyses on linezolid and fosfomycin have been already published [26,26,31].
- It would be worth supplementing the medications taken by patients. Were the patients taking anticoagulants? has there been any interaction?
Of note, meropenem is mainly excreted by the kidneys (filtration and active secretion), limiting the relevance of PK interactions due to e.g. hepatic metabolism (e.g. CYP-enzymes). Similarly, no dependence on a single isoenzyme responsible for its metabolism has been shown. Pre-operatively measured serum creatinine concentrations (mean, SD: obese 85.8 ± 26.3 µmol/L; non-obese: 75.3 ± 18.8 µmol/L) were in the normal range. Medication during surgery was similar in all patients (e.g. anaesthesia) or showed no detectable impact on MEM PK (e.g. vasopressors, data not shown).
The possibility of drug interactions has been considered, but no obvious interactions have been found. Due to the low sample size (per specific co-administered drug), interactions with a small effect size (probably clinically irrelevant anyway) may have been missed. We checked all the patients' medicines for possible interactions and found no effects on the pharmacokinetics of meropenem.
We have therefore added the following sentence to the results (page 3, lines 83-84):
No interactions between patients' medications were found to affect the pharmacokinetics of meropenem.
Additionally for you, here is the listing of all the patients' medication:
Table A. preoperative Comedikation
|
|
Obese (n=14/15) |
Non-obese (n=5/15) |
|
L-Thyroxin |
5 (33%) |
3 (20%) |
|
Amlodipin |
4 (27%) |
1 (7%) |
|
Hydrochlorothiazid |
4 (27%) |
1 (7%) |
|
Pantoprazol |
4 (27%) |
2 (10%) |
|
Torasemid |
4 (27%) |
0 |
|
Valsartan |
4 (27%) |
1 (7%) |
|
Metformin |
3 (20%) |
0 |
|
Metoprolol |
3 (20%) |
0 |
|
Ramipril |
3 (20%) |
0 |
|
Simvastatin |
3 (20%) |
0 |
|
Acetylsalicylsäure |
2 (13%) |
1 (7%) |
|
Bisoprolol |
2 (13%) |
0 |
|
Brinzolamid |
2 (13%) |
0 |
|
Candesartan |
2 (13%) |
0 |
|
Citalopram |
2 (13%) |
0 |
|
Duloxetin |
2 (13%) |
0 |
|
Formoterol |
2 (13%) |
0 |
|
Insulin |
2 (13%) |
0 |
|
Oxycodon |
2 (13%) |
0 |
|
Phenprocoumon |
2 (13%) |
0 |
|
Salbutamol |
2 (13%) |
0 |
|
Allopurinol |
1 (7%) |
0 |
|
Amiodaron |
1 (7%) |
0 |
|
Biotin |
1 (7%) |
0 |
|
Biperiden |
1 (7%) |
0 |
|
Chlorprothixen |
1 (7%) |
1 (7%) |
|
Diclofenac |
1 (7%) |
0 |
|
Eplerenon |
1 (7%) |
0 |
|
Exenatid |
1 (7%) |
0 |
|
Ezetimib |
1 (7%) |
0 |
|
Fluticason |
1 (7%) |
0 |
|
Folsäure |
1 (7%) |
1 (7%) |
|
Furosemid |
1 (7%) |
0 |
|
Glimepirid |
1 (7%) |
0 |
|
Ibubrofen |
1 (7%) |
0 |
|
Liraglutid |
1 (7%) |
0 |
|
Metamizol |
1 (7%) |
0 |
|
Multivitamine |
1 (7%) |
0 |
|
Nicotinamid |
1 (7%) |
0 |
|
Paroxetin |
1 (7%) |
0 |
|
Pregabalin |
1 (7%) |
1 (7%) |
|
Selen |
1 (7%) |
0 |
|
Theophyllin |
1 (7%) |
0 |
|
Tilidin |
1 (7%) |
0 |
|
Tiotropiumbromid |
1 (7%) |
0 |
|
Tramadol |
1 (7%) |
0 |
|
Travoprost |
1 (7%) |
0 |
|
Trospiumchlorid |
1 (7%) |
0 |
|
Urapidil |
1 (7%) |
1 (7%) |
|
Verapamil |
1 (7%) |
0 |
|
Zolpidem |
1 (7%) |
0 |
|
Pankreatin |
0 |
1 (7%) |
|
Pravastatin |
0 |
1 (7%) |
Entries are numbers (%).
- What are the weaknesses of the study? What are the limitations of the study?
The study had some limitations:
1.) The study was performed in non-infected patients. Infected patients, especially ICU patients may show a more variable PK. Extrapolation of our results to infected/ICU patients should be done with caution.
2.) PK may have been altered by the surgical situation (e.g. due to anaesthesia).
We mention these limitations at the end of the discussion and added a sentence regarding BMI (page 8, lines 192-196):
The BMI values in the obese group are very high and as such results may not be entirely applicable in non-bariatric cohorts. Moreover, by design, patients with BMI 30 to 34.9 kg/m2 were not included to have clearer separation between the groups. Since obese patients did not show any relevant differences to non-obese patients, it can be assumed that these limitations are minor.
- What's new in the presented study?
We have added following Sentence at the beginning of the discussion (page 7, lines 145-146):
The present study is the first clinical study with a direct comparison of meropenem tissue concentration in obese vs non-obese patients. The summarized findings of this study are that…
- Has there been a dose adjustment of the antibiotic in patients?
No. All patients received the same single dose as specified by the trial protocol and described in the Methods section. There are no recommendations for body weight (or BMI) adjusted dosages of MEM. Whether this common practice is reasonable or should be changed was the question of the present study.
We have made it more precise in Methods (page 9, lines 217-218):
As a perioperative antibiotic prophylaxis, patients were given a standard (weight independent) dose of 1000 mg meropenem (Meronem®, AstraZeneca GmbH, Wedel, Germany) in combination with 600 mg linezolid (Zyvoxid®, Pfizer Deutschland GmbH, Berlin, Germany) as a single 30-minute infusion after induction of anaesthesia (60-30 minutes prior to incision) through an additional vein access.
- What bacteria were identified in the blood and tissue cultures?
Meropenem was administered as a prophylaxis and no blood tests for pathogens were foreseen. Clinically, no infections were documented during the operation or within the week thereafter. There is following sentence on this issue in the results section in the first paragraph (page 2-3, lines 81-82):
No patient showed signs of acute infection or critical illness at the time of the surgery and the first 7 postoperative days.
- Has PK / PD been calculated separately for specific pathogens?
It was not the aim of the study to question the general efficacy of the widely used broad-spectrum antibiotic meropenem, but to detect differences between obese and non-obese patients. Meropenem is used against a broad spectrum of pathogens with various ECOFFs (e.g. Bacillus fragilis or Proteus mirabilis with 0.25 mg/L, Staphylococcus aureus with 0.5 mg/L, Ps. aeruginosa or Acinetobacter baumanii with 2 mg/L, etc.) in a variety of settings with varying targets (1xMIC, 4xMIC, etc.). We showed the time above 3 representative concentrations (T>MIC, low 0.25 mg/L, mid 2 mg/L, high 8 mg/L), which represents the PK/PD index for meropenem and demonstrated that T>MIC was not lower in obese compared to non-obese patients. However, and as already stated, the actual aim was to detect differences, which were small (clinically irrelevant) for all three threshold concentrations. In order to make Table 3 easier to read, we have deleted the unnecessary column "all patients" so that it is clearer what the aim of the study was: obese vs. non-obese (page 7).
- In the title of Figure 2, it should be stated that it is a logarithmic figure.
We have added this information to the caption (page 4, line 105).
Reviewer 2 Report
Paper by Doctor Simon P et al. compared serum concentration of antibiotics in obese and non-obese patients. I would like to provide comments to improve their manuscript.
[Major]
- Introduction: I would like to request the researchers to explain how dosage of antibiotics is determined for readers. Is it determined for people with standard BMIs?
- Methods: It would be natural that readers including me consider that if people with obesity has greater volume for their body, and that concentration of meropenem would be smaller in people with obesity than that in people within standard BMI. How do the researchers need to adjust the body size, or how do they discuss this influence?
- Table 3: The researchers could refer to the non-statistical significance in estimated difference in difference. Did they lack clinical significance, or is it due to small sample size?
- Figure 2 illustrates apparent difference between patients with obesity vs. standard BMI. This could be more greatly highlighted in the Discussion section. Is it possible that the researchers statistically detect that the two groups were divided on the figure 3? Could discriminant analysis perform the detection?
- Sample size: Is 15 patients in one arm enough to investigate this study question? I would like to request the appropriateness of this sample size.
- Discussion: Are the results specific to meropenem, or universal for all antibiotics or medicines? As a scientific paper, I would like to request the universality of the results in patients with obesity.
[Minor]
- I wonder how the researchers treated patients with BMI for 30.1 to 34.9 kg/m2. A volume zone of this BMI need to be discussed.
- Title: I wonder what was “Controlled” in this clinical trial. I recognise that this study is not adapted random assignment. What was controlled?
- Table 3: Positive figures in difference in difference had better to be added “+” in table 3.
Overall, sample size may be a key of this study value. This study could add evidence to the field of antibiotics.
Author Response
Point-by-point response to the reviewers’ comments to antibiotics-104004
“Meropenem plasma and interstitial soft tissue concentrations in obese
and non-obese patients – a controlled clinical trial”
by Simon et al.
Dear Prof. Dr. Nicholas Dixon, Editor-in-Chief,
Dear Dr. David P. Nicolau, Guest Editor,
Dear Leah Zhou,
Thank you very much for giving us the opportunity to revise our manuscript entitled „ Meropenem plasma and interstitial soft tissue concentrations in obese and non-obese patients – a controlled clinical trial”.
We thank the editors and reviewers for their criticism. We have discussed the points in detail and hope very much that after further revision you will come to the conclusion that the manuscript is suitable for publication in Antibiotics.
Our responses to the reviewers’ comments below are highlighted in blue and italic, our changes to the manuscript (insertions and deletions) are indicated in red (deletions are crossed through). The line numbers refer to those in the revised manuscript with highlighted changes.
REVIEWER 2:
Paper by Doctor Simon P et al. compared serum concentration of antibiotics in obese and non-obese patients. I would like to provide comments to improve their manuscript.
Thank you very much for your comments.
[Major]
- Introduction: I would like to request the researchers to explain how dosage of antibiotics is determined for readers. Is it determined for people with standard BMIs?
We have added the following sentence to the introduction (page 2, lines 58-59):
However, rrecommended daily doses of antibiotics are based on PK studies performed almost exclusively in non-obese patients [23].
- Methods: It would be natural that readers including me consider that if people with obesity has greater volume for their body, and that concentration of meropenem would be smaller in people with obesity than that in people within standard BMI. How do the researchers need to adjust the body size, or how do they discuss this influence?
There are no recommendations for body weight or BMI adjusted dosages of meropenem. Whether this common practice is reasonable or should be changed was the aim of the present study.
We have made it more precise in Methods (page 9, lines 217-218):
As a perioperative antibiotic prophylaxis, patients were given a standard (weight independent) dose of 1000 mg meropenem (Meronem®, AstraZeneca GmbH, Wedel, Germany) in combination with 600 mg linezolid (Zyvoxid®, Pfizer Deutschland GmbH, Berlin, Germany) as a single 30-minute infusion after induction of anaesthesia (60-30 minutes prior to incision) through an additional vein access.
- Table 3: The researchers could refer to the non-statistical significance in estimated difference in difference. Did they lack clinical significance, or is it due to small sample size?
There were no significant and no clinically relevant differences between obese and non-obese patients regarding T>MIC for MIC values 0.25, 2 and 8 mg/L. Of note, due to the longer half-life of MEM in obese patients the mean values in the obese group were even slightly higher in most cases (11.9 vs 10.2 h; 6.9 vs 6.1 h; 3.8 vs 4.2 h; 11.0 vs 9.6 h; 5.4 vs 5.0 h). Only for a MIC of 8 mg/L and only in ISF was the mean T>MIC slightly shorter in the obese group (1.4 h) compared to the non-obese group (2.0 h). Again, this difference was neither statistically significant nor clinically relevant.
We fully agree that estimates are more useful to an assessment of clinical relevance than a p-value, but in this case the estimates together with the confidence intervals suggest that the differences are indeed not relevant.
Regarding the clinical relevance and discussion of the sample size, we have added a sentence in the discussion (page 8, lines 175-181):
The decisive PK/PD parameter for beta-lactams is T>MIC. Plasma and tissue concentrations remain above the EUCAST breakpoints of 2 mg/L for Pseudomonas and Enterobacteriaceae [22] for comparably long periods of time suggesting that body weight adjusted dosage of meropenem in obesity is unnecessary after short-term infusion. In obese patients the mean values of T>MIC were somewhat higher in most cases though the difference is not clinically relevant, because a dose reduction for obese patients will not benefit them clinically. Although sample size was based on the primary endpoint, the width of the 95% CI suggests that it was also adequate for T>MIC.
- Figure 2 illustrates apparent difference between patients with obesity vs. standard BMI. This could be more greatly highlighted in the Discussion section.
The differences in Cmax and half-life between obese and non-obese patients in our study lead to a longer T>MIC in the obese group for MIC of up to 2 mg/L. Theoretically, a dose reduction in obese patients at a MIC of 2 mg/L would be conceivable. These are modest pharmacokinetic differences, but they have no effect on clinical routine, because a dose reduction for obese patients will not benefit them clinically. Therefore, we conclude that the results of our study do not suggest a body weight-adjusted dose adjustment after a short infusion of meropenem in obese patients (highlighted in discussion - page 8, lines 175-181).
Is it possible that the researchers statistically detect that the two groups were divided on the figure 3? Could discriminant analysis perform the detection?
The inclusion criteria for the groups were BMI < 30 kg/m2 and BMI > 35 kg/m2. BMI is so strongly associated with weight, that this leads to a perfect separation in Figure 3. Since this separation is by construction, a discriminant analysis would not provide additional information.
- Sample size: Is 15 patients in one arm enough to investigate this study question? I would like to request the appropriateness of this sample size.
Sample size calculation (as described in reference 25 of the manuscript: trial design) were based on the assumption that area-under-the-curve (AUC) for drug concentrations on a log-scale is roughly linear with weight. Simulations taking into account the expected weight distribution in the patients showed that 13 patients per treatment group have to be analysed to show a statistically significant difference in AUC with 80% power. To allow for drop-outs and missing data, 15 patients per weight category and drug combination were included.
We have added a sentence about sample size as it relates to T>MIC (page 8, lines 180-181):
Although sample size was based on the primary endpoint, the width of the 95% CI suggests that it was also adequate for T>MIC.
- Discussion: Are the results specific to meropenem, or universal for all antibiotics or medicines? As a scientific paper, I would like to request the universality of the results in patients with obesity.
Substances with similar physicochemical properties should behave similarly. Substances with comparable hydrophilicity/lipophilicity, with similar (i.e. with low or absent) protein binding and with similar molecular weight (etc.), which are mainly eliminated by the kidneys may behave similar. But such statements would be too hypothetical. Even within the same class of substances surprising differences can be found [e.g. ref. 8 in manuscript]. Usually, results assessed with a specific therapeutic, are only applied to this specific therapeutic.
We have added two supplementary sentences in the discussion (page 8, lines 164-166):
However, these effects of obesity are different for individual antibiotics [9], which is why these results are only valid for meropenem. For other substances (even quite similar ones, like doripenem) separate results are needed for evidence-based recommendations.
[Minor]
- I wonder how the researchers treated patients with BMI for 30.1 to 34.9 kg/m2. A volume zone of this BMI need to be discussed.
We added this sentence to this in the discussion (page 9, lines 192-196):
The BMI values in the obese group are very high and as such results may not be entirely applicable in non-bariatric cohorts. Moreover, by design, patients with BMI 30 to 34.9 kg/m2 were not included to have clearer separation between the groups. Since obese patients did not show any relevant differences to non-obese patients, it can be assumed that these limitations are minor.
- Title: I wonder what was “Controlled” in this clinical trial. I recognise that this study is not adapted random assignment. What was controlled?
We think it is a controlled study because the results in the study group (obese group) are compared with those in the control group (non-obese group). The term “controlled” (in the context of clinical trials) is not necessarily fixed to randomization. While the most known study type are certainly the “Randomized Controlled Trials” (RCTs) there are also (non-randomized) “Controlled Clinical Trials” (CCT). In our case, due to the aim and design of the study, randomization was not possible, of course. However, the present study is the first clinical study with a direct comparison of meropenem tissue concentration between obese and non-obese patients as a control group.
- Table 3: Positive figures in difference in difference had better to be added “+” in table 3.
We have adjusted it (page 7).
Round 2
Reviewer 2 Report
The researchers have addressed all of my comments. I have no more concern. I appreciate their efforts to report the important results.